# SCALABLE BATCH CORRECTION FOR CELL PAINTING VIA BATCH-DEPENDENT KERNELS AND ADAPTIVE SAMPLING

## ABSTRACT

Cell Painting is a microscopy-based, high-content imaging assay that captures rich morphological profiles of cells. By revealing how cells respond to different chemical perturbations, it can provide valuable insight for drug discovery. However, Cell Painting data suffers from batch effects caused by variations across laboratories, instruments, and protocols. These batch-dependent artifacts obscure biological signals, especially at scale. We introduce BALANS (read "balance")—Batch Alignment via Local Affinities and Subsampling—a scalable batch correction method that aligns samples across batches using a smoothing affinity matrix constructed based on pairwise distances between the data points. Given $n$ data points, BALANS constructs a sparse affinity matrix $A \in \mathbb{R}^{n \times n}$ following two key ideas. First, for data points $i$ and $j$, it defines a local "scale" based on the distance from $i$ to its $k$-th nearest neighbor within the batch of $j$. The affinities $A_{ij}$ are then computed using a Gaussian kernel calibrated by the local scales to account for batch-specific variation. Second, instead of populating all $n^2$ entries of $A$, BALANS employs an adaptive sampling strategy that incrementally computes rows corresponding to points with low cumulative neighbor coverage and, within each row, retains the highest affinities. This yields a sparse but informative submatrix of $A$. We prove that this novel sampling strategy is order-optimal in terms of sample complexity and has an approximation guarantee. Crucially, BALANS runs in almost-linear time with respect to the number of data points. We evaluate BALANS across many real-world datasets spanning diverse biological conditions and batch structures. We demonstrate scalability on these real-world datasets and perform controlled scalability experiments on large-scale synthetic data to assess efficiency under varying size and complexity. In both cases, BALANS outperforms native implementations of popular batch correction methods in runtime without compromising batch correction quality.

## 1 INTRODUCTION

Image-based profiling has emerged as a powerful tool for studying how cells respond to different treatments (Regev et al., 2017). Using high-throughput microscopy, researchers can detect changes in the shape and structure of cells caused by chemical or genetic perturbations (Rohban et al., 2017). This allows the identification of promising compounds based on the cellular effects they produce, offering a way to accelerate drug discovery. A widely used image-based profiling method is the Cell Painting assay (Bray et al., 2016; Caicedo et al., 2017). Originally developed by Bray et al. (2016), Cell Painting uses six fluorescent dyes to label eight distinct cellular components, which are imaged across four or five channels. This approach provides high-content, single-cell resolution data that is cost-effective (Cimini et al., 2023) and provides complementary information to (Cutiongco et al., 2020; Wawer et al., 2014) and protein-based assays (Dagher et al., 2023). Cell Painting has been successfully integrated with feature extraction techniques to generate numerical morphological profiles. A widely used tool is CellProfiler (Carpenter et al., 2006), which extracts profiles based on engineered features such as texture, intensity, and shape descriptors from segmented cellular components. More recently, deep learning-based methods like DeepProfiler (Moshkov et al., 2024) have been introduced, using architectures such as EfficientNet (Tan and Le, 2019) to extract profiles.

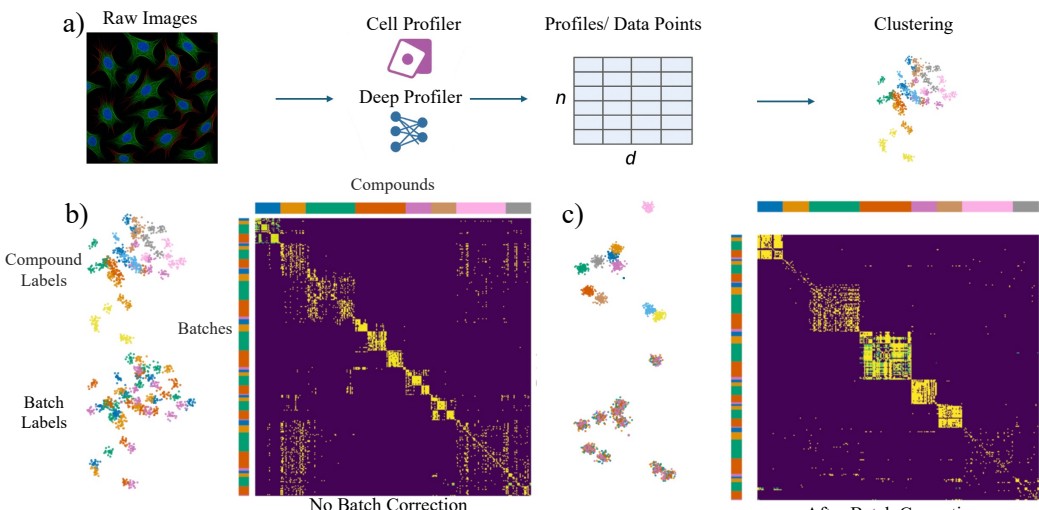

Figure 1: **(a)** Standard Cell Painting analysis pipeline based on extracting features for each of the $n$ cells. **(b)** Before correction: when UMAP plot is colored by compound labels (top), we see multiple clusters corresponding to same compound. When colored by batch, we see that the different clusters for the same compound correspond to different batches. The affinity heatmap reveals block patterns aligned with batches. **(c)** After BALANS: Clusters corresponding to different compounds sharpen, batches mix, and affinities align with compound labels. Heatmap rows are ordered by batch (left color bar), columns by label (top color bar).

**Batch effects obscure true biological signal:** Large-scale experiments often span multiple sources, (labs, protocols, microscopes) introducing batch-dependent artifacts that obscure true biological signals. The signal of interest is the morphological effect of a perturbation, independent of technical noise (e.g., plate or staining artifacts) or unrelated biological variation (e.g., cell density or nonspecific concentration effects). These confounding factors persist through standard pre-processing of the data. A comprehensive analysis of batch effects in Cell Painting data and the performance of existing batch correction techniques was provided by Arevalo et al. (2024).

To illustrate batch effects, we analyze real Cell Painting data using a curated set of positive control compounds from the JUMP-CP consortium (Chandrasekaran et al., 2023), which spans four datasets, 13 sources, and three microscopes. These compounds induce diverse morphological phenotypes and are thus ideal for visualizing batch effects. We used profiles extracted via CellProfiler The resulting affinity matrix is shown as a heatmap in Figure 1(b), prior to any batch correction, revealing a block structure aligned with batch labels (colors on the left of heatmap). The goal of batch correction is to perform a transformation on the cell feature vectors such that the resulting affinity matrix exhibits block structure that aligns with the compound labels, as illustrated in Figure 1(c).

**Affinity-Based Denoising:** Denoising methods are widely used to enhance biological signal by smoothing over similar data points (Van Dijk et al., 2018; Huang et al., 2018; Li and Li, 2018). They have shown success in mRNA and protein profiling (Tran et al., 2020). For instance, MAGIC (Van Dijk et al., 2018) constructs a carefully designed affinity matrix to average each point with its neighbors. We take inspiration from this denoising technique and seek to build *batch-aware* affinity matrices that can be used to remove batch effects. One challenge that arises is that computing all pairwise affinities can be computationally expensive when we have many data points, and Cell Painting experiments often scale to hundreds of thousands or even millions of single-cell profiles (Arevalo et al., 2024) across many different batches. Hence, we seek to design a *batch-aware* affinity matrix that can be used for batch correction in a *scalable* way, allowing the efficient batch correction of large datasets.

**BALANS:** In this work, we introduce BALANS—Batch Alignment via Local Affinities and Subsampling—a novel, scalable batch correction method. Unlike traditional affinity-based approaches, BALANS incorporates batch label information into the construction of the smoothing

affinity matrix. Scalability is attained by only computing a *sparse* version of the full affinity matrix, which allows each data point to be smoothed based on a small number of neighbors.

The first key idea behind BALANS is to make the notion of "closeness" depend on the batch. For any pair of data points $i$ and $j$ corresponding to unique profiles, BALANS adjusts their similarity by computing a local scale that depends on the batch of $j$. More precisely, if $b_j$ is the batch label associated with $j$, it computes the distance from $i$ to its $k$-th nearest neighbor within the batch of $j$, denoted $\sigma_{i,j} := d_{i,(k)}^{b_j}$, and uses this as a scaling factor in a Gaussian kernel. The affinity between $i$ and $j$ is then defined as $A_{ij} = \exp\left(-\|x_i - x_j\|^2/\sigma_{i,j}^2\right)$. This scaling has two effects. It increases the relative influence of compatible points from distant or less tightly clustered batches and suppresses misleading affinities from nearer or more tightly clustered batches. In doing so, BALANS selectively identifies cross-batch neighbors that are truly similar despite batch-induced distortions.

The second idea in BALANS is to avoid computing the full $n \times n$ affinity matrix, which is infeasible at scale. Instead, BALANS selects a subset $S \subseteq [n]$ with $|S| \ll n$ data points and computes their affinity rows. Rather than sampling uniformly, BALANS uses a coverage-based adaptive strategy: at each step $t$, the next point $i_t$ is selected with probability proportional to the inverse of its cumulative affinity to the previously sampled points. This prioritizes data points whose neighborhoods have not yet been well represented in earlier samples. Each computed affinity row is sparsified using an elbow detection method, retaining only the most informative entries. The resulting matrix $A_S \in \mathbb{R}^{|S| \times n}$ captures a partial view of the full affinity matrix $A \in \mathbb{R}^{n \times n}$. The remaining entries are inferred via a low-rank approximation based on $A_S$ (Williams and Seeger, 2000). BALANS is summarized in Figure 2 and Algorithm 1.

We theoretically prove that this sampling strategy guarantees a similar number of rows from each cluster with high probability, using only $|S| = O(K \log K)$ samples. This ensures coverage of the underlying structure, enabling a low rank approximation of $A$. We show that the reconstruction error $\|\widehat{A} - A\|_{\mathrm{op}}$ is bounded with high probability, where $\widehat{A}$ is constructed using the sampled rows. BALANS runs in *almost-linear time* with respect to the number of data points, making it scalable to large datasets. Importantly, the full affinity matrix $A$ is never explicitly constructed. Instead, BALANS applies a sparse, smoothing operator defined based on $A_S$ to the data matrix $X$.

**Experiments on Large Datasets:** We evaluate BALANS on real-world datasets from the JUMP Cell Painting Consortium (Chandrasekaran et al., 2023) and BBBC (Ljosa et al., 2012), with different scenarios using either CellProfiler or DeepProfiler features (Table 1). Across eight benchmark scenarios, BALANS performs on par with state-of-the-art batch correction methods, and in the largest-scale, highest-diversity setting (Table 1), it achieves over a 30% improvement in average evaluation scores relative to the uncorrected baseline. We also report runtime comparisons, showing that BALANS is fast and competitive with existing methods. In addition to accuracy, BALANS is highly efficient. Despite being implemented in Python, it runs faster than many optimized native implementations. On synthetic benchmarks designed to mimic real batch structure, while maintaining high correction quality, it scales to datasets with up to 5 million points in under an hour. These results highlight BALANS as a scalable and effective solution for batch correction in high-content imaging.

**Related Works:** Cell Painting has been extensively explored using deep learning-based feature representations beyond CellProfiler and DeepProfiler (Doron et al., 2023; Wong et al., 2023; Kraus et al., 2024). Batch correction has been extensively studied across a range of biological data modalities (Ando et al., 2017; Čuklina et al., 2021; Yu et al., 2023; Tran et al., 2020). Local bandwidth estimation (Zelnik-Manor and Perona, 2004; Herrmann, 1997; Knutsson et al., 1994) and low-rank approximation (Williams and Seeger, 2000; Drineas et al., 2005; Kumar et al., 2012; Musco and Musco, 2017) also have well-established foundations. A comprehensive review of related works is provided in Appendix L.

**Comparison to BBKNN:** BBKNN (Polański et al., 2020) is a tool that constructs a batch-aware KNN graph, and is related to BALANS. But its functionality differs fundamentally from BALANS. BBKNN produces only a graph structure and does not generate corrected profiles or embeddings. Although it defines an affinity matrix, BBKNN does not provide a way to efficiently apply this matrix to the data. BALANS, in contrast, is explicitly designed to output corrected profiles without ever forming the full matrix. Because BBKNN cannot produce corrected profiles, comparisons are limited to graph-based metrics. A more detailed discussion, along with quantitative results, is provided in the Appendix I.

## 2 KEY CONCEPTS

In Cell Painting assays, cells are subjected to various perturbations (e.g., chemical or genetic), imaged using fluorescence microscopy, and then processed through a feature extraction pipeline to obtain profiles representing cellular morphology. Consider $n$ data points $x_1, \ldots, x_n \in \mathbb{R}^d$ obtained in this way. Each point is annotated with a *known* batch label $b_i \in \{1, \ldots, B\}$, $i = 1, \ldots, n$ reflecting differences such as laboratory, protocols, or microscopes used. In addition, every point belongs to an *unknown* biological cluster. A true biological cluster is a group of points that share a common underlying phenotype or similar phenotypes, regardless of their treatment labels or batch assignments. Each point thus has a (unknown) cluster label $c_i \in \{\mathcal{C}_1, \mathcal{C}_2 \ldots, \mathcal{C}_K\}$, $i = 1, \ldots, n$. Our goal is to identify, for each data point, a small set of neighbors that reflect true biological similarity with the datapoint. Using its neighbors we are to construct a smoothing operator that corrects it.

To identify neighbors for smoothing, one relies on a similarity measure between samples. These similarities are commonly encoded in an affinity matrix, which can be constructed in various ways (kernel functions (Hofmann et al., 2008), nearest-neighbor graphs (Abbasifard et al., 2014), or learned distance metrics (Kulis et al., 2013)). However, when affinities are computed directly from the data without correction, they typically reflect both cluster membership and external sources of variation, including batch dependent effects and measurement noise. To account for batch-dependent distortions in pairwise similarities, we define a **batch-dependent local scale** for each point, capturing how distant each batch appears from its perspective. These scales are then used to compute affinities via a local-scale Gaussian Kernel (Zelnik-Manor and Perona, 2004), where the similarity between two points depends on this scale.

### 2.1 BATCH-DEPENDENT LOCAL SCALES

Let $d_{ij} = \|x_i - x_j\|_2$ denote the Euclidean distance between data points $x_i$ and $x_j$. A standard method for defining affinities from pairwise distances is using the Gaussian kernel $A_{ij} = \exp\left(-d_{ij}^2/\sigma^2\right)$, where $\sigma > 0$ is a global scale controlling the scale of locality. This kernel assigns higher affinity to closer points and is widely used in spectral clustering (Zelnik-Manor and Perona, 2004). However, the assumption of a fixed global scale $\sigma$ is problematic when different batches may exhibit distinct density patterns, noise levels or distances. Thus, this can lead to poor affinity estimates.

This issue can be observed directly on real data. Consider Figure 1(b), which shows a correlation matrix constructed as above, between data points that correspond to positive control compounds. Although these compounds are known to induce strong and diverse phenotypes, the resulting structure exhibits strong clustering within batches, which is undesirable. Therefore to address this issue, we define a batch-dependent local scale and the affinity associated with it.

**Definition 1** (Batch-dependent local scale; Affinity). Let $d_{ij} = \|x_i - x_j\|_2$ denote the Euclidean distance, and let $b_j$ be the batch of $x_j$. Let $\sigma_{ij}^2 := \left(d_{i,(k)}^{b_j}\right)^2$, where $d_{i,(k)}^{b_j}$ is the distance from $x_i$ to its $k$-th nearest neighbor within batch $b_j$. Then the batch-dependent affinity is $A_{ij} := \exp\left(-d_{ij}^2/\sigma_{ij}^2\right)$.

To illustrate its effect on real data, we apply the batch-dependent affinity to data points corresponding to positive controls and construct a correlation matrix. As shown in Figure 1(c), the resulting heatmap reveals improved structure, with clearer cross-batch alignment at a compound level. Computing batch-dependent affinities by adapting local scales across batches recovers true neighbors under batch effects, but is computationally expensive. It requires $O(n^2)$ distances and repeated $k$-NN searches to compute $\sigma_{ij}$ for all pairs. To ensure scalability, we estimate the affinity matrix by sampling a subset of rows, each of which provides access to a full neighborhood. From each row we obtain the batch-dependent local scale for that point. This form of row-based sampling is also known as landmark sampling (Kumar et al., 2012). Unlike randomly sampling entries of $A$, it preserves the information needed to compute batch-dependent scales. In the next section, we show that accurate approximation of $A$ is possible using only a small subset of rows.

### 2.2 LOW RANK MATRIX APPROXIMATION VIA ADAPTIVE LANDMARK SAMPLING

Before we proceed, we first require a suitable model on the structure of the affinity matrix. Going back to our experiment on positive controls, we note that applying the batch-dependent scale reveals

a block structure in the affinity matrix (Figure 1(c)), reflecting a block diagonal matrix overlaid with noise. This motivates a natural decomposition into a block diagonal matrix component plus noise. To formalize this, we model the observed affinity matrix $\widetilde{A} \in \mathbb{R}^{n \times n}$ as $\widetilde{A} = A_0 + E$, where $A_0$ is a block diagonal matrix encoding clustering structure, and $E$ denotes a residual noise independent of $A_0$. We assume $A_0$ is symmetric and low-rank, with the following structure.

**Assumption 2** (Block-structured affinities)**.** There exists a partition of the index set $[n]$ into true biological clusters $\mathcal{C}_1, \ldots, \mathcal{C}_K$, where each cluster $\mathcal{C}_k$ has size $n_k$. The low-rank affinity matrix $A_0 \in \mathbb{R}^{n \times n}$ then takes the block form $A_0 := \mathrm{diag}(A^{(1)}, A^{(2)}, \ldots, A^{(K)})$, where each diagonal block $A^{(k)} \in \mathbb{R}^{n_k \times n_k}$ represents the affinities within biological cluster $k$ and is given by $A^{(k)} = p_k \cdot \mathbf{1}_{n_k \times n_k}$, with $p_k > 0$ and $\mathbf{1}_{n_k \times n_k} \in \mathbb{R}^{n_k \times n_k}$ is the matrix of all ones. All off-diagonal blocks of $A_0$ are identically zero. The overall matrix $A_0$ is symmetric, positive semidefinite, and has rank $K$.

Note that while the off-diagonal blocks of $A_0$ are zero, this does not imply that there is no affinity between points in different clusters. Rather, the block-diagonal structure is a modeling choice for the purpose of selecting close data points with high similarities to the given data point.

We are still required to model the noise matrix $E$. A standard way to model noise is using Gaussian Orthogonal Ensembles (GOE) Mehta (2004), which are symmetric random matrices with independent Gaussian entries (up to symmetry) and are widely used to model noise. Importantly, their singular values concentrate around $O(\sqrt{n})$ with high probability, making them a useful model. However, in our setting, we are constrained to matrices with strictly positive entries, as $\tilde{A}$ is still a affinity matrix. We therefore adopt a randomized analogue with positive entries that retains the spectral scaling of GOEs.

**Assumption 3** (Exponential Noise Matrix)**.** Let $E \in \mathbb{R}^{n \times n}$ be a symmetric random matrix with $E_{ij} = E_{ji} \sim \mathrm{Exp}(\lambda)$ independently for all $i < j$, and $E_{ii} \sim \mathrm{Exp}(\lambda/2)$, for some $\lambda > 0$.

This ensures that $E$ is symmetric, and exhibits similar singular values as GOEs, while respecting entrywise positivity. Our goal is to recover $A_0$ using a small subset of rows from $\widetilde{A}$. To approximate $A_0$, we sample a subset $S \subseteq [n]$ of size $s$ and form the Nyström estimator (Kumar et al., 2012) $\widehat{A} := \widetilde{A}_S^\top \widetilde{A}_{S,S}^+ \widetilde{A}_S$, where $\widetilde{A}_S \in \mathbb{R}^{s \times n}$ contains the sampled rows and $\widetilde{A}_{S,S} \in \mathbb{R}^{s \times s}$ is their submatrix. Here, $A^+$ denotes the Moore–Penrose pseudoinverse (Courrieu, 2008), the unique matrix satisfying $AA^+A = A$ and $A^+AA^+ = A^+$. This yields a low-rank approximation of $A_0$ using the subspace defined by $S$.

How do we assess the quality of this estimator? The quality of reconstruction depends on how well the sampled rows capture connectivity across the data. Since each row of $\widetilde{A}$ represents affinities between a point and the rest of the dataset, a good sampling algorithm will sample informative rows across different regions of the data. To formalize this, we define a notion of *coverage*. Let $\mathrm{cov}(i)$ denote the cumulative affinity received by point $i$ from previously sampled rows, given by $\mathrm{cov}(i) := \sum_{j \in S_{\mathrm{past}}} \widetilde{A}_{ji}$, where $S_{\mathrm{past}}$ denotes the set of indices sampled so far. Intuitively, points with low cumulative coverage have not yet been well represented in the approximation. Sampling is performed in *blocks of size $K$*. At the start of each block, a temporary coverage vector is initialized, and the sampling distribution is uniform. At each step $t$ within the block, an index $i_t$ is drawn from a distribution $q^{(t)}$ that favors under-covered points, with $q_i^{(t)} \propto 1/\mathrm{cov}^{(t)}(i)$, where $\mathrm{cov}^{(t)}(i)$ is the accumulated coverage over the current block. After each block, the coverage vector is reset to $0$ and sampling distribution is reset to uniform. While Assumption 3 ensures that the affinity matrix $\widetilde{A}$ has strictly positive entries (so $\mathrm{cov}(i) > 0$ for all $i$), practical implementations involve sparsifying $\widetilde{A}$ (see Section 3), which can introduce zero coverage. In such cases, we modify the sampling rule as described in Algorithm 5. This alternating scheme ensures that sampling explores new regions of the dataset and targets under-covered areas. We analyze this novel sampling scheme and show that the estimator $\widehat{A}$ obtained from a small number of rows approximates $A_0$ with spectral guarantees.

### 2.3 THEORETICAL RESULTS

We now formalize the guarantees provided by coverage-based sampling for approximating the underlying affinity matrix $A_0$. Recall that the dataset is partitioned into $K$ biological clusters

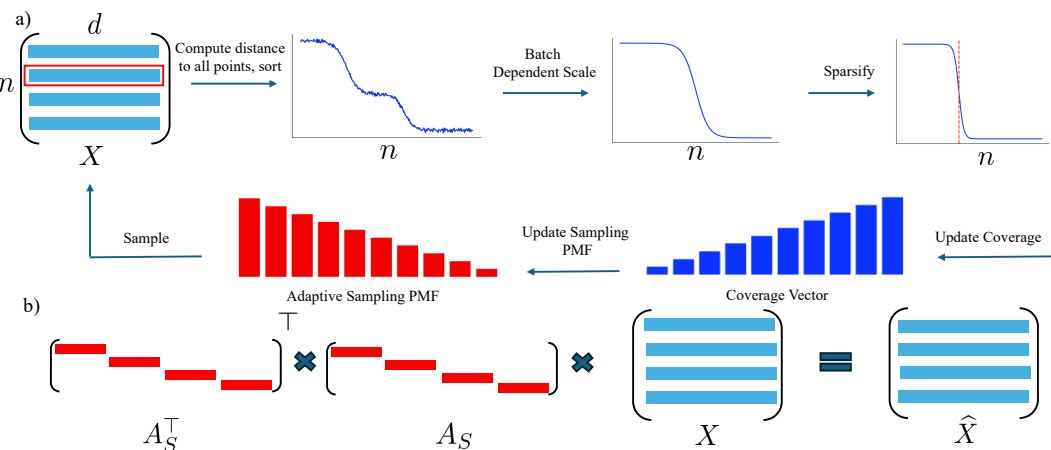

Figure 2: Overview of BALANS. **(a)** Adaptive sampling with batch-dependent scaling and sparsification. **(b)** Applying sparse smoothing to data.

$\mathcal{C}_1, \ldots, \mathcal{C}_K$ with sizes $n_1, \ldots, n_K$, and total size $n = \sum_{k=1}^{K} n_k$. Let $m$ be the number of rows sampled (We will discuss how $m$ is chosen through a stopping criteria in Section 3).

**Theorem 1** (Cluster Coverage Guarantee). *Let $T_k$ denote the number of sampled rows from $\mathcal{C}_k$. Then given a constant $\delta > 0$, there exists another constant $C := C(\delta)$ such that if $m \geq CtK \log K$, then $\mathbb{P}\left(T_k \geq t, \ \forall\ k\right) \geq 1 - \delta$.*

The proof (Appendix A) builds on a refined coupon collector argument (Boneh and Hofri, 1997). Unlike random sampling, the adaptive strategy achieves $\mathcal{O}(K \log K)$ coverage irrespective of cluster sizes, making it order-optimal. We also show in Appendix A that this property guarantees that our estimator $\widehat{A}$ closely approximates $A_0$ in operator norm. Let $\| \cdot \|_{\mathrm{op}}$ denote the operator norm.

**Theorem 2** (Spectral Approximation of $A_0$). *Let $\widehat{A} = \widetilde{A}_S^\top \widetilde{A}_{S,S}^+ \widetilde{A}_S$ denote the estimator formed by low-rank reconstruction from $m \geq C(\delta)tK \log K$ adaptively sampled rows of $\widetilde{A}$. Then,*

$$\|\widehat{A} - A_0\|_{\mathrm{op}} \leq \frac{Dn}{\sqrt{t} \cdot \min_k p_k}, \tag{1}$$

*with high probability, where $p_k$ is the cluster affinity for cluster $\mathcal{C}_k$, and $D := D(\lambda, K, \vec{p}) > 0$ depends on the parameter $\lambda$ of the noise matrix, the number of clusters $K$, and the affinities vector $\vec{p}$.*

Theorem 2 provides a guarantee on the spectral reconstruction error when enough rows are sampled. Specifically, when the number of samples $m = O(tK \log K)$, i.e. satisfying the condition in Theorem 1, using these samples suffices to achieve small spectral error that decays at the rate $O(1/\sqrt{t})$ while guaranteeing representation from all clusters.

## 3 THE BALANS ALGORITHM

Algorithm 1 and Figure 2 presents the full implementation of BALANS. Given a data matrix $X \in \mathbb{R}^{n \times d}$ and batch labels $\vec{b} \in \{1, \ldots, B\}^n$, the algorithm also takes as input a nearest-neighbor parameter $k$, a stopping threshold $\tau$, and an adaptive sampling parameter $J$. Optionally, the data may be projected onto a lower-dimensional PCA subspace. The algorithm maintains three variables: (i) a *cumulative coverage vector* $c \in \mathbb{R}^n$ tracking the cumulative affinity of each point (ii) a *temporary coverage vector* $c_K \in \mathbb{R}^n$ used to compute adaptive sampling probabilities over $J$ steps and (iii) a *set of sampled indices* $S$, initialized as empty. The main loop terminates when no new points are covered over $\tau$ consecutive iterations.

At each iteration, an index $i_t$ is sampled using the coverage-based adaptive strategy (Section 2.2, Alg. 5). For the selected point $x_{i_t}$, a sparse affinity row $A_{i_t,:}$ is computed using the batch-dependent local scale kernel (Section 2.1, Alg. 3). To reduce noise and memory usage, affinities are sparsified

via an elbow detection heuristic (Truong et al., 2020). Entries are first sorted in decreasing order, and only values above a sharp drop threshold are retained. After sparsification, $\Delta$ counts how many new indices receive nonzero affinity for the first time. If $\Delta = 0$, a stagnation counter is incremented; otherwise, it is reset. The loop stops after $\tau$ consecutive rounds with no new coverage, serving as a convergence heuristic. The collected affinity rows form a sparse matrix $A_S \in \mathbb{R}^{m \times n}$, which is compact and efficient to store.

In theory, the corrected data is computed via the low-rank estimator $\widehat{X} := (A_S^\top A_{S,S}^+ A_S)X$, where $A_S \in \mathbb{R}^{m \times n}$ contains sampled, row-normalized affinity rows, and $A_{S,S} \in \mathbb{R}^{m \times m}$ is the submatrix over sampled indices. This formulation enables efficient spectral reconstruction, as $A_S$ is sparse and $A_{S,S}^+$ is a small dense matrix. In practice, we omit $A_{S,S}^+$ to avoid its $O(m^3)$ cost, approximating the estimator as $\widehat{X} \approx (A_S^\top A_S)X$. Here, $A_S$ smooths $X$, and $A_S^\top$ propagates the result to all points. Both matrices are row-normalized for stability, and we avoid forming the dense matrix $A_S^\top A_S$ by computing $A_S X$ followed by $A_S^\top(A_S X)$. Since $A_S$ is sparse, all operations remain efficient.

---

**Algorithm 1** BALANS Batch Correction

---

1: **Input:** Data $X \in \mathbb{R}^{n \times d}$, batch labels $\vec{b} \in \{1, \ldots, B\}^n$, parameters $k, \tau, J$
2: **Output:** Corrected data $\widehat{X} \in \mathbb{R}^{n \times d}$
3: Compute PCA on $X$ (optional); set $c, c_K \leftarrow \mathbf{0}$; $S \leftarrow \emptyset$; no_change $\leftarrow 0$
4: **while** no_change $< \tau$ **do**
5:     **for** $j = 1$ to $J$ **do**
6:         Sample $i_t$ via Adaptive Sampling (Alg. 5); compute $A_{i_t,:}$ via Local Scale Kernel (Alg. 3)
7:         Append $A_{i_t,:}$ to $A_S$; update $S \leftarrow S \cup \{i_t\}$
8:         $\Delta \leftarrow \#\{j : A_{i_t,j} > 0 \text{ and } c_J[j] = 0\}$
9:         **if** $\Delta = 0$ **then**
10:             no_change_count $\leftarrow$ no_change_count $+ 1$
11:         **else**
12:             no_change_count $\leftarrow 0$
13:         **end if**
14:         $c \leftarrow c + A_{i_t,:}$;    $c_J \leftarrow c_J + A_{i_t,:}$
15:     **end for**
16:     Reset $c_J \leftarrow \mathbf{0}$
17: **end while**
18: Compute $\widehat{X}$ via Low-Rank Completion (Alg. 4); **return** $\widehat{X}$

---

Importantly, BALANS is robust to hyperparameters. Ablation studies are presented in Appendix E. We now state the overall computational complexity of BALANS shown in Appendix D.

**Theorem 3** (Computational Complexity of BALANS). *Let $n$ be the number of data points, $d$ the feature dimension, and $m = |S|$ the number of adaptively sampled rows after BALANS converges. Then the total computational complexity of* BALANS *is $\mathcal{O}(nm(d + \log n))$, where the first term accounts for sparse affinity construction and matrix multiplications, and the second term covers nearest neighbor search. When $m \ll n$, this is approximately linear in $n$.*

By Theorem 2, sampling $m = \mathcal{O}(K \log K)$ rows suffices to reconstruct $A_0$ with high probability while covering a large fraction of points, allowing the algorithm to terminate. This results in an overall computational complexity of $\mathcal{O}(nK \log K(d + \log n))$, which is near-linear in the dataset size $n$.

## 4 EXPERIMENTS

We evaluate BALANS on 8 real-world scenarios from 7 datasets, spanning both CellProfiler and DeepProfiler features. Five scenarios are from the JUMP Cell Painting Consortium (approximately 140,000 perturbations, 115 TB of images), and three use DeepProfiler embeddings from BBBC datasets (approximately 1000 perturbations with 50000 well averaged profiles). We also include 6 synthetic scenarios based on Gaussian mixtures ranging from 500 to 5 million data points. All methods are evaluated using metrics that jointly assess batch correction and biological signal preservation.

Full details are provided in Appendix F. We compare BALANS against a broad range of batch correction methods, including statistical approaches (Combat (Zhang et al., 2020), Sphering (Kessy et al., 2018)), linear projection methods (CCA, RPCA (Satija et al., 2015)), and graph-based or KNN-based methods (FastMNN , Scanorama (Hie et al., 2019), Harmony (Korsunsky et al., 2019)). These methods differ in their underlying assumptions. Table 3 in Appendix G summarizes these properties; full implementation details and references are provided in Appendix G.

We evaluate performance using a suite of established metrics (Luecken et al., 2022). These include Graph Connectivity, kBET, LISI (batch and label), Silhouette scores (batch and label), and clustering-based measures such as Leiden ARI and Leiden NMI. Each metric is normalized to the range [0, 1], with higher values indicating better performance. We note that kBET could not be computed for large-scale settings (e.g., JUMP 2) due to resource constraints. Full metric definitions and implementation details are provided in Appendix H. Table 4 in Appendix H provides a summary of the evaluation metrics. We highlight three scenarios here in Table 1 (full details in Appendix I). **JUMP 1**: medium-scale CellProfiler (25K points, 3 sources, 302 compounds). **JUMP 2**: large-scale, high-diversity (350K points, 5 microscopes, 80K+ compounds, low overlap). **DEEP 1**: DeepProfiler version of `BBBC036` with batch effects based on groups of plate rows. We also compare wall-clock runtimes (Table 2) for top-performing methods. BALANS outperforms other competing methods across datasets.

**Explaination of metrics:** We distinguish between batch metrics and bio metrics to provide a balanced view of integration quality. Batch metrics (e.g., Graph Connectivity, LISI-batch) assess how well spurious technical variation is removed, but can be misleading in isolation. Methods that erase all structure (e.g., by adding noise) may score highly despite discarding biological signal. Label metrics (e.g., ARI, NMI, Silhouette-label), in contrast, directly evaluate whether meaningful biological structure is preserved, and thus carry greater interpretive weight. Because no single metric captures all aspects, we emphasize comparative patterns and rely on Avg-all as a more robust summary across tradeoffs between local versus global structure. Ultimately, we prioritize improvements in label metrics, since recovering biologically coherent variation is the primary goal of batch correction. Notably, BALANS is consistently performs the best across all datasets on the average of label-based bio metrics (Avg-label), underscoring its ability to preserve biologically coherent variation. A detailed discussion on metrics is provided in Appendix F.

**Interpretation of metrics:** As an illustration of biological interpretability, BALANS more clearly recovers positive control compounds with known phenotypes. For example, the Aurora kinase inhibitor AMG900  forms a distinct cluster with a high silhouette score (0.82 vs. 0.45 for Harmony). This highlights BALANS's ability to preserve meaningful biological structure. Additional compound-level comparisons and heatmaps are provided in Appendix K.

BALANS performs the best on highly heterogeneous datasets, compared to baselines. In particular, the results on JUMP-1 and JUMP-2 illustrate this clearly. JUMP-2, which contains stronger batch effects and higher biological and technical heterogeneity, shows a clear advantage for BALANS across all metrics. On JUMP-1, where heterogeneity is more moderate, BALANS provides stable correction without substantially surpassing all baselines. Together, these results indicate that BALANS performs especially well as dataset heterogeneity increases, underscoring its suitability for large and diverse real-world settings.

To evaluate scalability, we introduce a synthetic benchmark based on a structured Gaussian mixture model. It supports datasets with up to 5 million points and varying batch complexity (details in Appendix J). The main experiment uses 10 compound clusters across 5 batches in 10 dimensions. As shown in Figure 4, BALANS achieves the fastest runtimes even on 5 million data points without compromising correction quality. Aggregate Scores for this experiment, additional scenarios, runtime details (including GPU hardware), and their aggregate scores are in Appendix J.

Table 1: Evaluation scores across three scenarios: **JUMP 2** (large, diverse batches), **DEEP 1** (low batch diversity, DeepProfiler), and **JUMP 1**. Since most methods are deterministic, the Standard Deviation (SD) are 0; we omit it here due to space. Tables with SDs are in the Appendix.

| Method | Conn. | LISI-batch | Silh-batch | LISI-label | ARI | NMI | Silh-label | Avg-batch | Avg-label | Avg-all |
|---|---|---|---|---|---|---|---|---|---|---|
| BALANS | 0.33 | **0.48** | **0.91** | **1.00** | 0.01 | **0.46** | 0.32 | **0.57** | **0.45** | **0.50** |
| Scanorama | **0.34** | 0.41 | 0.83 | **1.00** | 0.01 | 0.30 | 0.27 | 0.53 | 0.40 | 0.45 |
| SCVI | 0.28 | 0.44 | 0.82 | 0.99 | 0.01 | 0.28 | 0.29 | 0.53 | 0.39 | 0.44 |
| fastMNN | **0.34** | 0.45 | 0.79 | **1.00** | **0.02** | 0.26 | 0.22 | 0.53 | 0.38 | 0.44 |
| Harmony | **0.34** | 0.25 | 0.84 | **1.00** | 0.01 | 0.27 | 0.32 | 0.48 | 0.40 | 0.43 |
| Sphering | **0.34** | 0.00 | 0.84 | **1.00** | 0.00 | 0.22 | **0.38** | 0.39 | 0.40 | 0.40 |
| Combat | **0.34** | 0.02 | 0.81 | **1.00** | 0.00 | 0.24 | 0.31 | 0.39 | 0.39 | 0.39 |
| Baseline | **0.34** | 0.01 | 0.80 | **1.00** | 0.00 | 0.23 | 0.31 | 0.38 | 0.39 | 0.38 |
| DESC | 0.29 | 0.40 | 0.80 | 0.99 | 0.00 | 0.25 | 0.28 | 0.50 | 0.39 | 0.42 |

| Method | Conn. | LISI-batch | Silh-batch | LISI-label | ARI | NMI | Silh-label | Avg-batch | Avg-label | Avg-all |
|---|---|---|---|---|---|---|---|---|---|---|
| BALANS | 0.20 | 0.61 | 0.89 | **1.00** | **0.00** | 0.32 | 0.39 | **0.57** | 0.43 | **0.49** |
| Sphering | **0.30** | 0.43 | **0.90** | 0.99 | **0.00** | 0.30 | **0.46** | 0.54 | **0.44** | 0.48 |
| Seurat RPCA | 0.26 | **0.62** | 0.79 | 0.99 | **0.00** | 0.25 | 0.36 | 0.56 | 0.40 | 0.47 |
| Harmony | 0.26 | **0.62** | 0.80 | 0.99 | **0.00** | 0.23 | 0.37 | 0.56 | 0.40 | 0.47 |
| Seurat CCA | 0.27 | 0.60 | 0.79 | 0.99 | **0.00** | 0.27 | 0.36 | 0.55 | 0.40 | 0.47 |
| Baseline | 0.27 | 0.57 | 0.80 | 0.99 | **0.00** | 0.23 | 0.37 | 0.55 | 0.40 | 0.46 |
| Combat | 0.27 | 0.58 | 0.80 | 0.99 | **0.00** | 0.24 | 0.37 | 0.55 | 0.40 | 0.46 |
| fastMNN | 0.26 | 0.60 | 0.78 | 0.99 | **0.00** | 0.24 | 0.35 | 0.55 | 0.40 | 0.46 |
| Scanorama | 0.19 | 0.56 | 0.80 | **1.00** | **0.00** | **0.34** | 0.33 | 0.52 | 0.42 | 0.46 |
| SCVI | 0.24 | 0.58 | 0.82 | 0.99 | **0.00** | 0.29 | 0.34 | 0.55 | 0.41 | 0.47 |
| DESC | 0.23 | 0.55 | 0.79 | 0.99 | **0.00** | 0.26 | 0.33 | 0.53 | 0.40 | 0.46 |

| Method | Conn. | KBET | LISI-batch | Silh-batch | LISI$_l$ | ARI | NMI | Silh-label | Avg-batch | Avg-label | Avg-all |
|---|---|---|---|---|---|---|---|---|---|---|---|
| Seurat CCA | 0.59 | **0.61** | 0.50 | 0.88 | **0.98** | **0.05** | 0.40 | 0.47 | **0.65** | 0.48 | **0.56** |
| BALANS | 0.54 | 0.45 | 0.44 | **0.89** | **0.98** | **0.05** | **0.46** | **0.53** | 0.58 | **0.51** | 0.54 |
| Seurat RPCA | 0.59 | 0.46 | 0.41 | 0.88 | **0.98** | 0.03 | 0.39 | 0.47 | 0.59 | 0.47 | 0.53 |
| fastMNN | 0.53 | **0.61** | 0.46 | 0.83 | 0.97 | 0.03 | 0.35 | 0.43 | 0.61 | 0.45 | 0.53 |
| Harmony | **0.60** | 0.41 | 0.43 | 0.88 | **0.98** | 0.03 | 0.39 | 0.47 | 0.58 | 0.47 | 0.52 |
| Scanorama | 0.35 | 0.56 | **0.52** | 0.80 | **0.98** | 0.03 | 0.34 | 0.43 | 0.56 | 0.45 | 0.50 |
| SCVI | 0.50 | 0.40 | 0.45 | 0.85 | 0.97 | 0.03 | 0.38 | 0.46 | 0.58 | 0.47 | 0.51 |
| Baseline | 0.51 | 0.17 | 0.13 | 0.78 | **0.98** | 0.02 | 0.32 | 0.47 | 0.40 | 0.45 | 0.42 |
| Combat | 0.54 | 0.08 | 0.11 | 0.80 | **0.98** | 0.02 | 0.33 | 0.47 | 0.38 | 0.45 | 0.42 |
| Sphering | 0.48 | 0.10 | 0.06 | 0.79 | **0.98** | 0.01 | 0.28 | 0.47 | 0.36 | 0.44 | 0.40 |
| DESC | 0.49 | 0.35 | 0.42 | 0.84 | 0.97 | 0.02 | 0.36 | 0.45 | 0.57 | 0.46 | 0.50 |

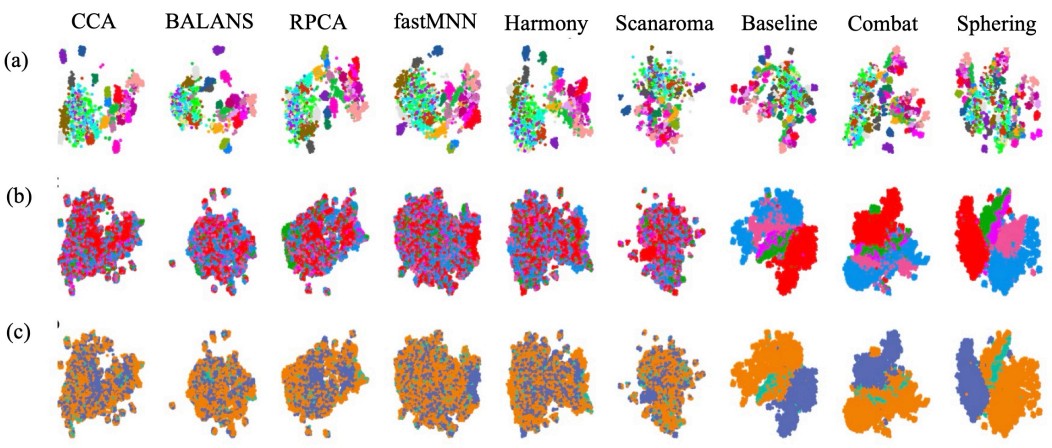

Figure 3: Figure 3: UMAP visualizations for JUMP 1. (a) Colored by biological cluster, illustrating separation across a few representative compounds. (b) Colored by source lab. (c) Colored by microscope. These panels demonstrate how batch effects manifest across different technical sources. Notably, BALANS yields a tight and coherent biological clusters while substantially reducing separation across labs and microscopes, indicating strong batch mixing without sacrificing biological structure.

Table 2: Wall-clock runtimes (in minutes) for best performing methods across real-world scenarios.

| Method | Scenario JUMP 1 | Scenario JUMP 2 | Scenario DEEP 1 |
|---|---|---|---|
| BALANS | **00:00:59** | **00:55:30** | **00:00:44** |
| Harmony | 00:12:35 | 01:44:53 | 00:22:43 |
| Scanorama | 00:01:33 | 01:13:02 | 00:01:57 |
| Seurat CCA ** | 00:34:37 | 12:00:00+ | 01:03:24 |
| Seurat RPCA ** | 00:14:03 | 12:00:00+ | 00:21:29 |

Figure 4: Runtime comparison on synthetic data (10 compounds, 5 batches, 10 features). Left: all methods. Right: fastest methods only. X-axis: number of data points (log scale); Y-axis: runtime in seconds. BALANS shows excellent scalability and maintains competitive runtime even at large dataset sizes, highlighting its suitability for large-scale batch correction tasks.

## 5 CONCLUDING REMARKS

We introduce BALANS, a novel method that integrates local affinity structure with adaptive sampling to provide a scalable and theoretically grounded approach to batch correction in Cell Painting data. Empirically, BALANS exceeds baseline performance by over 30% on the most diverse and large-scale dataset evaluated, and is shown to be faster than competing high-performing methods.

## 6 REPRODUCIBILITY STATEMENT

All code, data-processing scripts, and experimental pipelines associated with this work will be made publicly available on GitHub. Detailed instructions will be provided to enable other researchers to reproduce our results end-to-end, including installation, dataset preparation, and execution steps. Furthermore, all theoretical assumptions underlying our methods are explicitly described in the main paper to ensure clarity and transparency.

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
