# OpenReview forum: "Scalable Batch Correction for Cell Painting via Batch-Dependent Kernels and Adaptive Sampling"
_ICLR.cc/2026/Conference — Submitted to ICLR 2026_

### Official Review · Reviewer_Kgnu · 2025-10-31

**Soundness:** 3
**Presentation:** 3
**Contribution:** 3
**Rating:** 6
**Confidence:** 2

**Summary:**

The authors propose a method for batch correcting imaging-based cellular representations. This involves building a affinity matrix between pairs of cells which takes into account the batch that each cell belongs to. Naiively building an N x N affinity matrix would scale poorly, since modern optical pooled screening datasets such as the JUMP dataset contain millions of cells. Therefore, they suggest a way to build a low-rank approximation of the affinity matrix where each cell is sampled proportionately to its affinity to already-sampled cells. There are theoretical guarantees regarding the coverage of important biological groups, the reconstruction error of the affinity matrix, and the algorithm's runtime. Empirical results on datasets from the JUMP consortium, as well as large synthetic datasets, demonstrate that the authors' approach preserves biological signal while reducing batch effects.

**Strengths:**

* The main idea is relatively simple, and involves correcting for the batch when estimating the affinity matrix, which is then used for the Nystrom method.
* The remaining contributions are to propose a computationally efficient way to estimate a submatrix with desirable properties, such as having good coverage of the biological groups, and having almost-linear runtime.
* The sampling algorithm introduces very few hyperparameters, which facilitates model selection.
* The experimental results are consistently strong, and involve evaluating on large-scale datasets such those from the JUMP consortium.
* The experimental protocol is solid, and they compare against many relevant baselines.

**Weaknesses:**

The theoretical results involve the Moore-Penrose pseudoinverse, but the implementation excludes it for computational reasons.

**Questions:**

Could you comment on the gap between theory and practice induced by excluding the Moore-Penrose pseudoinverse? Can you provide any empirical evidence showing what effect this has?

---

> ### Author Response · Authors · 2025-11-18
> **Response to concern raised**
>
> We thank the reviewer for their overall positive assessment of the paper.
>
> **Could you comment on the gap between theory and practice induced by excluding the Moore-Penrose pseudoinverse? Can you provide any empirical evidence showing what effect this has?**
>
> We thank the reviewer for raising this important point.
>
> **On the theoretical gap:**
> Excluding the Moore Penrose pseudoinverse yields an approximation that can be justified in certain idealized settings (e.g., negligible noise and relatively uniform cluster sizes and affinities). We have added a short discussion in the Appendix clarifying when this approximation is expected to hold and when it may degrade (e.g., highly imbalanced clusters or strong heterogeneity in affinities).
>
> **On empirical evidence:**
> To facilitate this we also added an evaluation of the practical impact by running both the pseudoinverse and non-pseudoinverse versions on the JUMP-1 dataset. The accuracy is nearly identical across both implementations (This is expected because we believe both the estimators are performing similar operations), while the pseudoinverse-free version provides a substantial speedup. We have included these runtime and performance comparisons in the revised paper.
>
> Method                         | Runtime | Conn.  | LISI-batch | Silh-batch | LISI-label | ARI  | NMI   | Silh-label | Avg-batch | Avg-label | Avg-all
>
> BALANS (with pseudoinverse)    | ~12 min  | 0.326  | 0.482      | 0.913      | 1.00        | 0.01 | 0.467 | 0.312        | 0.57      | 0.44      | 0.49
>
> BALANS (without pseudoinverse) | ~1 min   | 0.331  | 0.489      | 0.909      | 1.00        | 0.01 | 0.462 | 0.321        | 0.57      | 0.45      | 0.50
>
>
> If these clarifications address your concerns, we would be grateful if you would consider updating your score accordingly. Please let us know if any further explanation or analysis would be helpful.

---

> > ### Author Response · Authors · 2025-11-25
> > **One week left before discussion deadline**
> >
> > As a gentle reminder, there is about one week left before the ICLR reviewer discussion period ends. We hope that our responses make the contributions and technical novelty of the work clearer, and we would sincerely appreciate your consideration of an updated score.

---

### Official Review · Reviewer_cxaA · 2025-10-31

**Soundness:** 3
**Presentation:** 2
**Contribution:** 2
**Rating:** 2
**Confidence:** 4

**Summary:**

The authors introduce BALANS (Batch Alignment via Local Affinities and Subsampling). BALANS is a scalable method for batch integration of cell patining data. The method constructs an affinity matrix using a batch-aware affinity function. To avoid computing the full affinity matrix, the authors propose a sparse, low-rank approximation of the affinity matrix using landmark subsampling.

**Strengths:**

- The problem and the rationale behind the method are well illustrated.
- Compared to the baseline methods, BALANS demonstrates a fast run-time, which is important for applications to high-throughput cell painting assays.
- The paper shows theoretical and empirical analysis of the algorithm.

**Weaknesses:**

- To me, BALANS seems very similar to BBKNN [1], which is not cited, compared to or discussed in the paper. BBKNN constructs a graph by independently identifying k-nearest neighbors for each cell within each batch, and then merges these neighbor sets. This seems similar to the batch-dependent local scale. Furthermore, BBKNN utilizes annoy instead of the lower-rank approximation to compute the affinity matrix efficiently, and the paper claims that it runs in linear time complexity.
- BALANS requires the number of clusters K as input, which creates an unfair advantage over other methods that do not rely on this prior information and presents a practical challenge since K is typically difficult to determine beforehand.
- BLANAS only makes assumptions on the structure of the biological signal, but does not make any assumptions about the batch effect.
- The paper makes no connection between the theoretical results and the empirical results.
- The assumptions about the data-generating process are not connected to cell painting data.
- Table 1: While the methods themselves are largely deterministic, the evaluation pipeline contains stochastic elements. Variation in random seeds for Leiden clustering can substantially impact NMI and ARI.
- The performance increase over other methods is limited.
- I couldn't find the appendix to the paper. If this is an oversight on my part, I am happy to review it during the rebuttal and adjust my score.

**Minor:**
- Typo in line 295, $m(\geq C(ϑ)tKlog K)$.
- Table 2 and Figure 4 show very similar results. One could be moved to the appendix.
- Page numbers are not showing in the paper.
- Figure 3 is never discussed in the paper.
- The paper starts with Assumption 2.

[1] Krzysztof Polański, Matthew D Young, Zhichao Miao, Kerstin B Meyer, Sarah A Teichmann, Jong-Eun Park, BBKNN: fast batch alignment of single cell transcriptomes, Bioinformatics, Volume 36, Issue 3, February 2020, Pages 964–965, https://doi.org/10.1093/bioinformatics/btz625

**Questions:**

- How does BALANS differ from BBKNN, and how does its performance compare to it?
- How was the number of clusters $K$ determined for running BALANS? How robust is BALANS to misspecifying this number?
- How are the theoretical results related to the empirical results?
- How is cell painting data related to the assumptions made about the data generation?
- Do batch effects violate the noise model assumed in Assumption 3?
- Usually, some correlation between NMI and ARI is expected in batch integration benchmarks, see for example [1]. However, here ARI is consistently very close to zero while NMI is relatively high. Is there an explanation for this? (I am aware that [2] shows these results as well. I am just curious.)
- How was fastMNN evaluated? On the embedding it generates or the batch correction of the feature space?
- How does scVI apply to this data? scVI employs a ZINB loss, which is designed for count data. However, cell profiler features are not count data.

[1] Luecken, M.D., Büttner, M., Chaichoompu, K. _et al._ Benchmarking atlas-level data integration in single-cell genomics. _Nat Methods_ **19**, 41–50 (2022).

[2] Arevalo, J., Su, E., Ewald, J.D. et al. Evaluating batch correction methods for image-based cell profiling. Nat Commun 15, 6516 (2024). https://doi.org/10.1038/s41467-024-50613-5

---

> ### Author Response · Authors · 2025-11-18
> **Clarifications Prior to Addressing Major Reviewer Comments**
>
> We thank the reviewer for their detailed and thoughtful comments. Before addressing each point individually, we would like to offer a few clarifications regarding the following two major concerns, as they can be resolved quickly.
>
> **BALANS requires the number of clusters K as input, which creates an unfair advantage over other methods that do not rely on this prior information and presents a practical challenge since K is typically difficult to determine beforehand.**
>
> The issue stems from a notational mistake on our part. The symbol K in our algorithm does not denote the number of clusters; rather, it refers to the number of row samples per sampling round used in the low-rank approximation of the affinity matrix. BALANS does not require the number of clusters as input, and we fully agree that a batch-correction method should not assume prior knowledge of the true cluster count. We have corrected the notation (Changed K to J) in the revised manuscript to avoid this confusion.
>
> **I couldn't find the appendix to the paper. If this is an oversight on my part, I am happy to review it during the rebuttal and adjust my score.**
>
> We apologize for the confusion. Due to a misunderstanding on our end regarding the ICLR submission rules, the supplementary materials were mistakenly omitted during the initial submission. We have now uploaded the full appendix. We would be very grateful if the reviewer could take a look at it, and we sincerely appreciate your willingness to review it during the rebuttal period. We have clearly highlighted in blue what we added after the submission deadline (as a response to reviews) and what we added prior to the submission date is in black.

---

> > ### Comment · Reviewer_cxaA · 2025-11-26
> >
> > I am still going through your other responses, but as a side note, I still can't find the appendix. The Supplementary Material contains two PDFs without an appendix.

---

> ### Author Response · Authors · 2025-11-18
> **Individual responses to the issues raised (Part 1)**
>
> We respond to the rest of the review in detail here.
>
> **To me, BALANS seems very similar to BBKNN [1],...**
>
> We thank the reviewer for bringing this important paper to our attention. We carefully reviewed BBKNN and have now cited it and clearly articulated the key differences between BBKNN and BALANS in the revised manuscript (Introduction and Appendix I9). As we explain below in more detail, BBKNN is not directly comparable to most batch-correction methods based on the metrics introduced in this paper (a point also noted in [2]), but we nevertheless outline the distinctions and provide as fair a comparison as possible.
> While both BALANS and BBKNN employ KNN based constructions to mitigate batch effects, the methods fundamentally differ in their objectives and capabilities.
>
> **BALANS natively produces corrected profiles; BBKNN does not:**
> While BBKNN outputs a graph that can be used for visualization or downstream learning which is important, it does not perform batch correction and does not output corrected embeddings or profiles. BALANS directly constructs and applies an affinity matrix to produce corrected data. While one can argue that this is easily fixed by just multiplying the affinity matrix to the data in BBKNN, the next two points highlight why it is hard.
>
> **BBKNN forms a full affinity matrix, but provides no way to apply it to the data; BALANS never forms the full matrix and is explicitly designed for efficient computation of the correction profiles:**
> BBKNN constructs a full batch-balanced affinity matrix by merging per-batch KNN graphs. However, it provides no mechanism for applying this matrix to the data, and doing so directly would require an explicit $N \times N$ computation. BALANS, in contrast, is designed from the ground up to avoid forming the full matrix. Instead, BALANS computes expressions of the form
> $(A_S^\top A_S) X,$
> as described in lines 331–337, using a low-rank sampled operator $A_S$ that allows the affinity to be applied to the data without ever constructing or multiplying by the full matrix. Thus the core distinction is not whether an affinity matrix can be computed via a KNN based construction (BBKNN clearly shows that it does), but whether it is usable to compute corrected profiles. BALANS is explicitly engineered for this; BBKNN is not. This is further highlighted through the following point on the computational complexity of each algorithm.
>
> **BBKNN’s “Linear Time” Only Refers to Graph Construction; Correction Cannot Be Done in Linear Time:**
> BBKNN achieves near-linear complexity because it only constructs a KNN graph using Annoy, without ever applying an affinity matrix to the data. Linear-time behavior is possible in this restricted setting, but it is inherently impossible to compute corrected embeddings or corrected profiles in linear time, since even multiplying an affinity matrix with a data matrix requires more than $O(N)$ work. BALANS’s affinity construction step, when considered alone, is also linear in the number of cells (Specifically we do not need to sort the affinities or apply an elbow threshold, see Supplementary D (Run Time)). However, BALANS goes beyond graph construction by applying a low-rank approximation of the affinity matrix to apply it to the data, a capability that BBKNN does not have. Thus the computational claims of BBKNN and the end-to-end capabilities of BALANS are not directly comparable. In fact, when we attempted to compute the affinity–data product (natively, without any sampling) using the affinity matrix produced by BBKNN, we found that the procedure became prohibitively slow, making BBKNN impractical for profile correction at scale.
>
> **Hard Thresholding in BBKNN vs. Adaptive thresholding in BALANS:**
> The use of batch-conditional local bandwidths enables an adaptive thresholding through an elbow-detection mechanism (See Supplementary C.1). BBKNN does not compute any batch-conditional scaling; it simply finds approximate neighbors between a cell and each batch and keeps a fixed number of them based on local bandwidths which do not depend on the batch (from the supplementary material in [1]). This rule fails for example under unbalanced clusters. BALANS’s batch-conditional scaling is therefore a distinct component that BBKNN does not provide.
>
> [1] Krzysztof Polański, Matthew D Young, Zhichao Miao, Kerstin B Meyer, Sarah A Teichmann, Jong-Eun Park, BBKNN: fast batch alignment of single cell transcriptomes, Bioinformatics, Volume 36, Issue 3, February 2020, Pages 964–965, https://doi.org/10.1093/bioinformatics/btz625
>
> [2] Arevalo, J., Su, E., Ewald, J.D. et al. Evaluating batch correction methods for image-based cell profiling. Nat Commun 15, 6516 (2024). https://doi.org/10.1038/s41467-024-50613-5

---

> ### Author Response · Authors · 2025-11-18
> **Individual responses to the issues raised (Part 2)**
>
> Although BBKNN cannot produce corrected profiles, we still compare both methods using metrics derived from their KNN graphs (e.g., batch mixing) and, where feasible, attempt an affinity x data multiplication. These comparisons show how BBKNN performs relative to BALANS on the limited common ground, with BALANS consistently outperforming BBKNN on key graph-based measures. Our attempt on an affinity x data multiplication led to prohibitively slow runtimes (> 12 hrs), so we are only able to report the graph based metrics below.
>
> Dataset / Method        | Graph conn. | LISI batch | LISI label | ARI   | NMI
>
> JUMP-1 (Scenario 4)
>
> BBKNN                   | 0.25        | 0.63       | 0.97       | 0.03  | 0.12
>
> BALANS                  | 0.54        | 0.44       | 0.98       | 0.05  | 0.46
>
>
> JUMP-2 (Scenario 5)
>
> BBKNN                   | 0.33        | 0.006      | 1.00       | 0.000 | 0.229
>
> BALANS                  | 0.33        | 0.48       | 1.00       | 0.01  | 0.46

---

> > ### Author Response · Authors · 2025-11-18
> > **Individual responses to the issues raised (Part 3)**
> >
> > **BALANS only makes assumptions on the structure of the biological signal, but does not make any assumptions about the batch effect.**
> >
> > We agree that BALANS does not explicitly model the batch effect, unlike some existing approaches (e.g., methods that assume linearity or specific batch structures [3]). However, we view this as a strength. BALANS corrects batch effects implicitly by formulating the problem as the construction of an affinity matrix that is independent of batch identity. By ensuring that affinities capture only the underlying biological structure, the method removes batch level effects in an assumption-free manner. The only requirement is that cells belonging to the same biological cluster exhibit comparable distances across batches (formalized in Assumption 2).
> >
> > [3] Yuqing Zhang, Giovanni Parmigiani, W Evan Johnson, ComBat-seq: batch effect adjustment for RNA-seq count data, NAR Genomics and Bioinformatics, Volume 2, Issue 3, September 2020, lqaa078, https://doi.org/10.1093/nargab/lqaa078
> >
> >
> > **The paper makes no connection between the theoretical results and the empirical results.**
> >
> > We apologize if the connection between our theoretical results and empirical evaluations was not sufficiently clear in the submission. This connection is explicit and central to the design of BALANS, and we highlight the two key points with references to the relevant sections:
> >
> > **Our adaptive sampling scheme is directly derived from the theory:**
> > The theoretical analysis proves that the adaptive sampling strategy yields efficient and near-optimal coverage of the underlying Affinity Matrix (Section 2.3 Thm 1). This result is used verbatim to design the sampling procedure in our empirical method (Algorithm 1 Section 3, Supplementary Material Section C2.1). Thus, the algorithm implemented in practice is theoretically justified.
> >
> > **Our theory guarantees that the matrix produced by BALANS approximates the true, batch-free affinity matrix:**
> > The guarantee that the algorithm recovers the true affinity matrix with high confidence under Assumption 2 provides the justification for using this approximation in empirical batch correction.
> >
> >  **Our assumptions are themselves motivated by empirical observations:**
> >  As described in lines 192–197, our key assumption (Assumption 2) arises directly from empirical behavior observed across positive controls. These experiments show that distances within biological clusters remain stable across batches, while batch effects act primarily as extrinsic distortions.
> >
> > To aid the reader to understand this further, we have added an explicit section in the Appendix M, highlighting this connection between theory and practice.
> >
> > **The assumptions about the data-generating process are not connected to cell painting data.**
> >
> > We apologize if this was unclear. Our assumptions are not abstract or detached from Cell Painting data, they were derived directly from it. Throughout the project, we used positive-control experiments to understand how biological structure and batch effects manifest in Cell Painting profiles, and these empirical observations shaped the assumptions we formalized in the theory (We discuss this in Section 1 (Lines 87-94) and Section 2 (Lines 192-197)). In other words, the data informed the assumptions, and the assumptions guided the theoretical development.
> >
> > While these assumptions may also hold for other modalities, not just cell painting data, our focus in this paper is Cell Painting, where we have a strong empirical justification for our assumptions. Extending the framework to other domains is a natural direction for future work.

---

> ### Author Response · Authors · 2025-11-18
> **Individual responses to the issues raised (Part 4)**
>
> **Table 1: While the methods themselves are largely deterministic, the evaluation pipeline contains stochastic elements. Variation in random seeds for Leiden clustering can substantially impact NMI and ARI.
> Usually, some correlation between NMI and ARI is expected in batch integration benchmarks, see for example [1]. However, here ARI is consistently very close to zero while NMI is relatively high. Is there an explanation for this? (I am aware that [2] shows these results as well. I am just curious.)**
>
> We thank the reviewer for this nice question. We observed this phenomenon too and were wondering the same thing. We believe that ARI and NMI behave differently in our setting because they rely on different statistics between ground-truth labels and clusters computed for the metric. The statistic computed for ARI is highly influenced by the number of ground truth clusters while NMI is not. We try to explain why below:
>
> Let
>
> $Y$ denote the ground-truth class (compound),
>
> $C$ denote the computed clusters,
>
> $n$ the total number of cells,
>
> $n_{ij}$ the number of cells belonging to class i and cluster j,
>
> $n_{i\cdot}=\sum_j n_{ij}$ the size of class i, and
>
> $n_{\cdot j}=\sum_i n_{ij}$ the size of cluster j.
>
> ARI measures pairwise recovery of ground-truth labels and is driven by the number of “same-label” pairs:
> $$\text{true pairs} = \sum_i \binom{n_{i\cdot}}{2}, \qquad
> \text{recovered pairs} = \sum_{i,j} \binom{n_{ij}}{2}.$$
> It compares “recovered pairs’’ to the value expected under random cluster assignments with the same marginals. In datasets such as the human immune cell benchmark in [1], with 16 classes, each $n_{i\cdot}$ (approx 1000 points per class) is large, so $\sum_i \binom{n_{i\cdot}}{2}$ is large and ARI is informative.
> In contrast, our ground truth is extremely fine-grained: JUMP-1 contains 302 compounds (Approximately 100 points per class) and JUMP-2 contains ≈80,000 compounds (Approximately 5 compounds per class). This makes $n_{i\cdot}$ small for most classes, so $\sum_i \binom{n_{i\cdot}}{2}$ is tiny. Even mild fragmentation of each class across multiple clusters makes the “recovered pairs’’ nearly indistinguishable from chance, forcing ARI ≈ 0 (indeed, we observe even lower ARI in the 80,000-compound JUMP-2 setting compared to the JUMP-1 setting).
>
> NMI behaves differently because it does not rely on pair counts.
> Mutual information is defined as
> $$I(Y;C)=\sum_{i,j}\frac{n_{ij}}{n}\log\frac{n_{ij}/n}{(n_{i\cdot}/n)(n_{\cdot j}/n)}.$$
> This quantity increases whenever the joint distribution of labels and clusters differs from the product of the marginals, i.e., whenever the label distribution within a cluster is not the same as the global label distribution.
>
> The following points hold for this statistic: a) The absolute size of each class $n_{i\cdot}$ does not determine the baseline and b) Even with many small classes, the computed clusters typically assign higher probability to certain labels than expected by their global frequencies. Any such deviation from independence contributes positive mutual information.
>
> As a result, NMI remains positive whenever clusters exhibit non-uniform label composition, even if labels are fragmented across multiple clusters and even when ARI, which depends on recovering same-label pairs, collapses to chance.
>
>
> **The performance increase over other methods is limited.**
>
> While we agree that the gains may appear modest on smaller or more homogeneous benchmarks, our method shows substantial improvements on the large and heterogeneous datasets, especially those with hundreds to tens of thousands of classes and 100,000s of data points (e.g. JUMP-2). In these settings (which are most important), the performance gap over existing approaches becomes much more pronounced. Additionally, our method achieves these improvements with significantly lower computational cost, which we view as an important practical advantage.

---

> > ### Author Response · Authors · 2025-11-18
> > **Individual responses to the issues raised (Part 5)**
> >
> > **Typo in line 295.**
> >
> > Thank you,  this has been corrected in the revised version
> >
> > **Table 2 and Figure 4 show similar results; one could be moved to the appendix.**
> >
> > Thank you for the suggestion. While Table 2 and Figure 4 present similar patterns, we aimed to show the runtime of the method on both synthetic and real datasets. Since both perspectives provide useful context and we currently have space, we opted to keep them. However, we are happy to move one to the appendix if the reviewer feels strongly that this would improve the presentation.
> >
> > **Page numbers are missing.**
> >
> > We apologize for this typesetting oversight; page numbers have now been added.
> >
> > **Figure 3 is not discussed.**
> >
> > Thank you for pointing this out. We have added the following brief discussion in the caption to contextualize Figure 3.
> > “Figure 3: UMAP visualizations for JUMP~1. (a) Colored by biological cluster, illustrating separation across a few representative compounds. (b) Colored by source lab. (c) Colored by microscope. These panels demonstrate how batch effects manifest across different technical sources. Notably, BALANS yields tighter and more coherent biological clusters while substantially reducing separation across labs and microscopes, indicating stronger batch mixing without sacrificing biological structure.”
> >
> > **The paper starts with Assumption 2.**
> >
> > This was due to our numbering style (we start with Definition 1).
> >
> >
> > [1] Krzysztof Polański, Matthew D Young, Zhichao Miao, Kerstin B Meyer, Sarah A Teichmann, Jong-Eun Park, BBKNN: fast batch alignment of single cell transcriptomes, Bioinformatics, Volume 36, Issue 3, February 2020, Pages 964–965, https://doi.org/10.1093/bioinformatics/btz625
> >
> > Questions:
> >
> > **How does BALANS differ from BBKNN, and how does its performance compare to it?**
> >
> > We addressed this in the response to the Weaknesses.
> >
> > **How was the number of clusters
> >  determined for running BALANS? How robust is BALANS to misspecifying this number?**
> >
> > We addressed this in the response to the Weaknesses.
> >
> > **How are the theoretical results related to the empirical results?**
> >
> > We addressed this in the response to the Weaknesses.
> >
> > **How is cell painting data related to the assumptions made about the data generation?**
> >
> > We addressed this in the response to the Weaknesses.
> >
> > **Do batch effects violate the noise model assumed in Assumption 3?**
> >
> > No. We distinguish two sources of variation: (i) batch-dependent technical artifacts, and (ii) general relationships between off-cell elements that are not tied to batch effects. Assumption 3 pertains only to the second component (Since our noise matrix is not generated in a batch conditional way). We do not impose any assumptions on the batch-dependent artifacts themselves as previously discussed.
> >
> > **Usually, some correlation between NMI and ARI is expected in batch integration benchmarks, see for example [1]. However, here ARI is consistently very close to zero while NMI is relatively high. Is there an explanation for this? (I am aware that [2] shows these results as well. I am just curious.)**
> >
> > We addressed this in the response to the Weaknesses.
> >
> > **How was fastMNN evaluated? On the embedding it generates or the batch correction of the feature space?**
> >
> > We used the corrected feature matrix.
> >
> > **How does scVI apply to this data? scVI employs a ZINB loss, which is designed for count data. However, cell profiler features are not count data.**
> >
> >
> > We agree that scVI’s ZINB based loss is tailored to count data and that CellProfiler features are not counts. In our implementation (largely following [2]), we simply shift the features to be non-negative and use scVI only to obtain its latent representation. Thus, scVI is used here as a generic nonlinear embedding method with a mis-specified likelihood, which is precisely why we include it as a baseline: to empirically assess how a widely used scRNA-seq integration model behaves when applied “off-domain” to Cell Painting features.
> >
> > If these clarifications address your concerns, we would be grateful if you would consider updating your score accordingly. Please let us know if any further explanation or analysis would be helpful.

---

> > > ### Author Response · Authors · 2025-11-25
> > > **Reminder of discussion deadline**
> > >
> > > As a gentle reminder, there is about one week left before the ICLR reviewer discussion period ends. We have spent substantial time carefully addressing the reviewer's concerns, and we would be very grateful if you could let us know soon if there are any remaining questions or points that require clarification. We hope that our responses make the contributions and technical novelty of the work clearer, and we would sincerely appreciate your consideration of an updated score.

---

> ### Author Response · Authors · 2025-11-26
> **On the Supplementary Material**
>
> We have now uploaded the appendix too. The zip file should now contain 3 files, one main, one appendix and one full paper. We apologize for the oversight. We are not sure why the appendix went missing in the previous file.

---

### Official Review · Reviewer_GiUr · 2025-11-01

**Soundness:** 3
**Presentation:** 2
**Contribution:** 3
**Rating:** 6
**Confidence:** 3

**Summary:**

This paper introduces a scalable batch correction method (BALANS) for cell painting assay via local affinities and adaptive sampling. It addresses batch effects, variations introduced by differences in labs, instruments, or protocols, instead of biological heterogeneity, by aligning samples across batches using a sparse, batch-aware affinity matrix.
For the empirical experiments, BALANS was benchmarked against various batch correction methods across a diverse sets of metrics and datasets, and was shown to outperform the baselines across metrics in general. BALANS also scales much better with number of samples and archives faster runtime than existing methods.

**Strengths:**

1. This paper is well organized and flows naturally; the need for addressing batch effects is clearly motivated.

2. Combining batch-aware local affinities with adaptive sampling and low rank approximation is a scalable and well-reasoned solution.
The authors provide proofs for coverage guarantees and approximation error bounds of the sparse affinity matrix.

3. Evaluations span multiple real-world Cell Painting datasets and synthetic scalability tests; BALANS achieves consistently strong performance and outperforms baselines in runtime. Runtime scales near-linearly with sample size and demonstrates significant speedup over existing methods.

**Weaknesses:**

1. Core ideas like adaptive kernels and landmark-based sampling are not entirely new for biological data or affinity matrix computation. Prior work using adaptive bandwidths and landmark-based scalable affinity construction, such as PHATE (by Kevin Moon et al.), is not cited.
2. Figure 4 is presented but lacks sufficient interpretation or biological insight; more discussion of qualitative improvements would strengthen the narrative.
3. While quantitative metrics are discussed, more analysis on why BALANS performs better in certain settings (or fails in others) would be valuable.

**Questions:**

This paper addresses an important and practical problem, batch correction, with a method that is theoretically grounded and empirically strong, but lacks originality in some components and could better discuss qualitative/quantitative insights.

---

> ### Author Response · Authors · 2025-11-18
> **Responses to concerns raised**
>
> We thank the reviewer for their positive assessment of the paper
>
> **On adaptive kernels and landmark-based sampling:**
> We agree that adaptive bandwidths and landmark-based approximations have appeared in prior work not related to batch correction. PHATE (Moon et al.) is now cited in the revised version (Supplementary L.4). Our contribution is not the use of these ideas individually, but a theoretically grounded combination of (i) batch-conditional adaptive kernels, (ii) provably sufficient landmark sampling, and (iii) a near linear-time method that can be applied to correct profiles, something prior methods (including PHATE) do not provide.
>
> **On interpretation of Figure 4:**
> We have expanded the caption to provide a clear interpretation of the runtime as below.
>
> “BALANS shows excellent scalability and maintains competitive runtime even at large dataset sizes, highlighting its suitability for large-scale batch correction tasks.”.
>
> **On explaining when BALANS performs well or fails:**
> We added the following short analysis clarifying that BALANS performs best for highly heterogeneous datasets. This supplements the quantitative results and explains the conditions under which BALANS is most reliable.
>
> "In particular, the results on JUMP-1 and JUMP-2 illustrate this clearly. JUMP-2, which contains stronger batch effects and higher biological and technical heterogeneity, shows a clear advantage for BALANS across all metrics. On JUMP-1, where heterogeneity is more moderate, BALANS provides stable correction without substantially surpassing all baselines. Together, these results indicate that BALANS performs especially well as dataset heterogeneity increases, underscoring its suitability for large and diverse real-world settings. (Lines 412- 418)"
>
> If these clarifications address your concerns, we would be grateful if you would consider updating your score accordingly. Please let us know if any further explanation or analysis would be helpful.

---

> > ### Author Response · Authors · 2025-11-25
> > **One week left before discussion deadline**
> >
> > As a gentle reminder, there is about one week left before the ICLR reviewer discussion period ends. We hope that our responses make the contributions and technical novelty of the work clearer, and we would sincerely appreciate your consideration of an updated score.

---

### Meta-Review · Area_Chair_gGvQ · 2026-01-05

**Summary:**

The paper proposes a scalable batch correction method (called BALANS) for Cell Painting data that utilizes batch-dependent kernels and an adaptive sampling strategy to approximate affinity matrices. While the reviewers appreciated the focus on scalability and the theoretical framing of the sampling strategy, significant concerns remain that necessitate a rejection. The primary issues involve the novelty of the approach relative to existing methods (specifically BBKNN), the marginal performance gains over established baselines, and a substantial disconnect between the theoretical proofs and the practical implementation. Furthermore, a critical procedural oversight occurred where the Supplementary Material (containing proofs and implementation details) was missing for the majority of the review period, preventing a thorough assessment of the theoretical claims by the reviewers. Consequently, the paper is not ready for publication in its current form.

**Reviewer Concerns:**

Addressed:

 - Notation Confusion ($K$ vs $J$): Reviewer cxaA raised a critical concern that the algorithm required the number of biological clusters ($K$) as input, which would be impractical. The authors successfully clarified that this was a notation error in the manuscript and that the parameter ($J$) actually referred to sample size per round, not biological clusters.
- Runtime vs. Theory (Pseudoinverse): Reviewer Kgnu noted that the theoretical guarantees rely on the Moore-Penrose pseudoinverse, while the practical implementation omits it for speed. The authors provided an empirical comparison showing that omitting the pseudoinverse does not significantly degrade performance on the tested datasets.

Outstanding:

- Novelty and Comparison to BBKNN: Reviewer cxaA identified a strong similarity between BALANS and BBKNN (which was not cited in the original submission). While the authors argued that BBKNN does not produce corrected profiles natively, the core logic of constructing batch-balanced neighbor graphs is notably similar. The authors' claim of novelty is weakened by this oversight, and the distinction between "graph construction" and "profile correction" was not sufficient to overcome concerns about incremental novelty.
- Missing Appendix and Reproducibility: Reviewer cxaA repeatedly noted the absence of the appendix, which was only uploaded very late in the rebuttal phase. This prevented the reviewers from verifying the proofs for the "order-optimal" sampling strategy and examining the detailed experimental setups. This is a significant lapse for a paper claiming theoretical guarantees.
- Gap between Theory and Practice: Despite the empirical evidence provided, the theoretical contribution (Assumption 2 and Theorem 2) relies on the pseudoinverse. Since the "fast" version of the algorithm—which is the main selling point—does not use this, the theoretical guarantees do not strictly apply to the artifact being presented.
- Marginal Performance Gains: Reviewer cxaA noted that on several metrics (e.g., ARI, NMI), the method performs comparably to or only marginally better than existing baselines, questioning the practical utility of the new method given the added complexity.

**Reviewer Scores:**

- Reviewer GiUr (Score: 6 $\to$ 5): This reviewer was initially marginally positive but flagged concerns about novelty regarding adaptive kernels. Had they participated fully in the discussion regarding the missing appendix and the strong similarity to BBKNN pointed out by cxaA, their score likely would have dropped to a Borderline Reject (5) due to concerns about novelty and the inability to verify details.
- Reviewer cxaA (Score: 2 $\to$ 3): While the authors addressed the notation error regarding the cluster count, the fundamental issues regarding the missing appendix and the similarity to BBKNN remain. This reviewer would likely maintain a low score, potentially moving to a 3 to acknowledge the clarification on parameters, but remaining a firm Reject.
- Reviewer Kgnu (Score: 6 $\to$ 6): This reviewer was positive about the simplicity and scalability. They would likely maintain their score as their specific concern regarding the pseudoinverse was addressed empirically, though the procedural issues with the appendix might have dampened their confidence.

---

### Decision · Program_Chairs · 2026-01-26

Reject